# *Campylobacter* occurrence and antimicrobial resistance profile in under five-year-old diarrheal children, backyard farm animals, and companion pets

**Wondemagegn Mulu[1,2], Marie Joossens[1], Mulugeta Kibret[3], Anne-Marie Van den Abeele[4,5], Kurt Houf [1,5]***

1 Laboratory of Microbiology, Department of Biochemistry and Microbiology, Ghent University, Ghent, Belgium, 2 Department of Medical Laboratory Science, College of Medicine and Health Sciences, Bahir Dar University, Bahir Dar, Ethiopia, 3 Department of Biology, Science College, Bahir Dar University, Bahir Dar, Ethiopia, 4 Laboratory of Microbiology, Sint-Lucas Hospital, Ghent, Belgium, 5 Department of Veterinary and Biosciences, Faculty of Veterinary Medicine, Ghent University, Ghent, Belgium

\* kurt.houf@ugent.be

**Data Availability Statement:** All relevant data are within the paper and its Supporting Information files.

## Abstract

Campylobacteriosis disproportionately affects children under five in low-income countries. However, epidemiological and antimicrobial resistance (AMR) information at the children-animal interface is lacking. We hypothesized that *Campylobacter* is a major cause of enteritis in children in Ethiopia, and contact with animals is a potential source of transmission. The objective of the study was to determine *Campylobacter* occurrence and its AMR in children under five with diarrhea, backyard farm animals, and companion pets. Stool from 303 children and feces from 711 animals were sampled. *Campylobacter* was isolated through membrane filtration on modified charcoal cefoperazone deoxycholate agar plates under microaerobic incubation, and the technique showed to be feasible for use in regions lacking organized laboratories. Typical isolates were characterized with MALDI-TOF MS and multiplex PCR. Of 303 children, 20% (n = 59) were infected, with a higher proportion in the 6 to 11-month age group. *Campylobacter* occurred in 64% (n = 14) of dogs and 44% (n = 112) of poultry. *Campylobacter jejuni* was present in both a child and animal species in 15% (n = 23) of 149 households positive for *Campylobacter*. MICs using the gradient strip diffusion test of 128 isolates displayed resistance rates of 20% to ciprofloxacin and 11% to doxycycline. MICs of ciprofloxacin and doxycycline varied between *C. coli* and *C. jejuni*, with higher resistance in *C. coli* and poultry isolates. C*ampylobacter* infection in children and its prevalent excretion from backyard poultry and dogs is a understudied concern. The co-occurrence of *C. jejuni* in animals and children suggest household-level transmission As resistance to ciprofloxacin and doxycycline was observed, therapy of severe campylobacteriosis should consider susceptibility testing. Findings from this study can support evidence-based diagnosis, antimicrobial treatment, and further investigations on the spread of AMR mechanisms for informed One Health intervention.

**Funding:** This work was supported by a research fund of Bahir Dar University, Bahir Dar, Ethiopia (Research grant to WM), and by the Research Council of Ghent university, Belgium (DOS-BOF 01W03820 to WM). The funders had no role in study design, data collection and analysis, decision to publish, or preparation of the manuscript.

## Author summary

Diarrheal diseases are the second-leading cause of death in children under five years, and *Campylobacter* is a major global causative agent. However, little attention is given to the accurate isolation, characterization, and AMR assessment of *Campylobacter* in sub-Saharan Africa. This region has the highest burden of diarrhea but a lack of organized laboratories. This study investigated the presence and AMR of *Campylobacter* in children and animals using methods with minimal technical challenge.

We collected stool samples from 303 children at health centers and 711 animal feces samples from the children's homes in the vicinity of Bahir Dar, Ethiopia. *Campylobacter* were isolated using membrane filtration-culture. Presumptive isolates were characterized with molecular techniques in Belgium. Membrane filtration culture with a traditional microaerobic incubation system was successful in isolating viable *Campylobacter* from children with diarrhea and animals. *Campylobacter* is highly prevalent in children, backyard poultry, and dogs and has shown resistance to ciprofloxacin and doxycycline. The key findings of this study clearly show that *Campylobacter* infection in children and its occurrence in animals associated with children is a major public health problem. These findings are supportive of clinical decision-making and treatment. Findings from this study are also relevant as a validation of applying the membrane filtration-isolation technique for *Campylobacter* screening in low-income countries.

## Introduction

Gastrointestinal diseases cause around 525,000 deaths yearly in children under five [1] and *Campylobacter jejuni* and *Campylobacter coli* have been ranked as one of the four principal microorganisms causing gastroenteritis worldwide [2,3]. Campylobacteriosis occurs in all age groups in high-income countries, but it is widespread and disproportionately affects children under five in low-income countries [4]. A pooled prevalence of 10% was reported in children under five, and 9% in a general population in Ethiopia [5,6]. In children under five, it is strongly linked to severe morbidity, environmental enteric dysfunction, malnutrition, failure to thrive, neurological disorders and death [7]. Children afflicted with campylobacteriosis experience fever, vomiting, abdominal cramps, watery to severe inflammatory diarrhea and dehydration [8]. Furthermore, post-infection complications of campylobacteriosis include arthritis, Guillain-Barré syndrome resulting in respiratory and neurological dysfunction, and irritable bowel syndrome [9].

Ciprofloxacin is one of the first choice treatment options in severe cases of bacterial enteritis [10]. Currently, children are however mostly treated without prior identification of the specific causative organism and antimicrobial susceptibility profiling in Ethiopia [11]. The use of inappropriate antimicrobials for empirical treatment might contribute to the selection of resistant strains and AMR. Antimicrobial resistance can result from a genetic mutation within the organism or acquiring resistance genes from other organisms following selection pressures in humans and animals [12].

*Campylobacter* can colonize, persist, and exert different pathogenicity in a variety of hosts, and is considered the main cause of foodborne bacterial zoonotic infections [13]. Poultry, pigs, and cattle are important reservoirs for the thermophilic *Campylobacter* species, *C. jejuni* and *C. coli* [14,15]. Direct contact with feces of poultry, ruminants, and pets, as well as consumption of contaminated food of animal origin and drinking water, are considered the main transmission routes [16,17]. Due to the broad host range and uncontrolled uses of antimicrobials in

human and veterinary medicine, and agriculture, the emergence and dissemination of AMR in *Campylobacter* is both a global health challenge and threat to One Health [18–20].

*Campylobacter* is a Gram-negative curved or spiral, motile, microaerobic and fastidious bacterium, and not easy to cultivate in sub-optimal laboratory conditions, as in some parts of Ethiopia [2,3]. As a result, insufficient attention is given to the epidemiological burden of disease and to optimal isolation and correct identification of *Campylobacter* from children with diarrhea. Because of variability in isolation procedures with respect to the choice of culture media and systems of microaerobic condition, temperature, and duration of incubation, results from previous studies are inconsistent. Aside from two recent studies, Terefe et al.[7] and Chala et al.[20], which applied molecular methods to detect *Campylobacter* in stool, the majority of Ethiopian studies have been solely performed by classical and biochemical identification methods. Both studies using molecular tools, included limited numbers of samples (less than 100), and each was carried out in a single study center [7,20]. Also, studies involving humans, poultry, and/or cattle from the same households are rare, participate small sample sizes (total number of human and animal samples combined: n = 347 and 70) [20,21], and describe individuals of not age specific [20] and infants [21], regardless of the status of their diarrhea.

*Campylobacter* exposure to antimicrobials is expected to be high. Inappropriate empirical treatment of enteritis in children is common in the study area, and resistant infections result in toxicity, increased morbidity and mortality rates, and a heightened cost of care [12]. Up till now, data on *Campylobacter* enteritis and antimicrobial susceptibility testing (AST) profiles in children with acute diarrhea in homes with backyard farm animals and/or companion pets is expected to be inaccurate in Ethiopia, due to the use of difficult to compare and/or suboptimal methods for isolation and AST.

Around 85% of rural households in Ethiopia own cattle, sheep, goats, poultry, and pets which interact closely with family members in the residences [22]. They are frequently kept indoors and roam freely so that animal feces are present in-house. As a result, young children can acquire *Campylobacter* directly from animals or food, water and surfaces contaminated with animal waste. However, knowledge on the occurrence of *Campylobacter* between children under five and animals is lacking in Ethiopia. Due to the lack of toilet facilities, human defecation in the open, close to homes and drinking water sources, is also a common practice. Therefore, assessing the presence of *Campylobacter* in children and animals, along with their susceptibility towards locally available antimicrobials, is highly relevant.

Consequently, the objectives of the present study were to determine the occurrence of *Campylobacter* in a large cohort of children under the age of five years with diarrhea, as well as in backyard farm animals and companion pets associated with these children. Moreover, the study aimed to determine the AMR profiles of *Campylobacter* isolates from children and animals to antimicrobials locally prescribed or available and recommended by standard guideline [23]. Based on these objectives, we conducted this study to test our hypothesis that due to environmental conditions, *Campylobacter* is a major cause of enteritis among children under five years old in Ethiopia, with contact with backyard farm animals are the primary source of transmission in the vicinity of Bahir Dar, northwest Ethiopia. New methodologies were used in order to generate baseline data that could support clinical decision-making and treatment, and provide insights on *Campylobacter* transmission and AMR between children and animals for the overall mitigation of campylobacteriosis and AMR.

## Materials and methods

### Ethics statement

The Institutional Review Board of the College of Medicine and Health Sciences (CMHS) of Bahir Dar University reviewed and approved the study protocols with reference number

CMHS/IRB 01–008. A formal letter of support was obtained from Bahir Dar University CMHS and Amhara Public Health Institute. Data and samples were collected after receiving written informed consent from the child's parents. Those formal consent Health centers were reached through the Bahir Dar City and Bahir Dar Zuria Health Departments. All survey, observation, and laboratory analysis data were anonymized, numbered, coded, and kept strictly confidential. For the proper management of patients, *Campylobacter* positive results were conveyed to nurses working at the health centers. Members of the households were informed about good practices in sanitation and hygiene to prevent *Campylobacter* infection and diarrhea. After authorization from the Ethiopian Food and Drug Authority, colonies presumptive of *Campylobacter* were transported from Bahir Dar, Ethiopia to Ghent, Belgium following national and international specimen transfer rules and regulations.

## Study participants

Children under five suffering from diarrhea, who sought treatment at the health centers involved, and from whom sufficient feces for sample collection could be collected, were eligible to participate. Children were eligible to participate upon written parental consent. Children experiencing diarrhea for more than seven days and coming from a household without any backyard farm animals were excluded.

## Study area and design

A healthcare facility-based cross-sectional study was conducted in the surroundings of Bahir Dar City, Ethiopia, from March 15, 2021, to April 15, 2022. Bahir Dar City is located at the point where the Blue Nile River leaves Lake Tana and at an altitude of 1820 m (GPS coordinates of 11˚ 34' 27.1524" N and 37˚ 21' 40.8708" E) (Fig 1A) [24]. Children with diarrhea were recruited from Kinbaba, Robit, Wonjeta and Yinesa Health centers in Bahir Dar Zuria, and from Abay, Han and Zege Health Centers in Bahir Dar City (Fig 1B). Based on the data

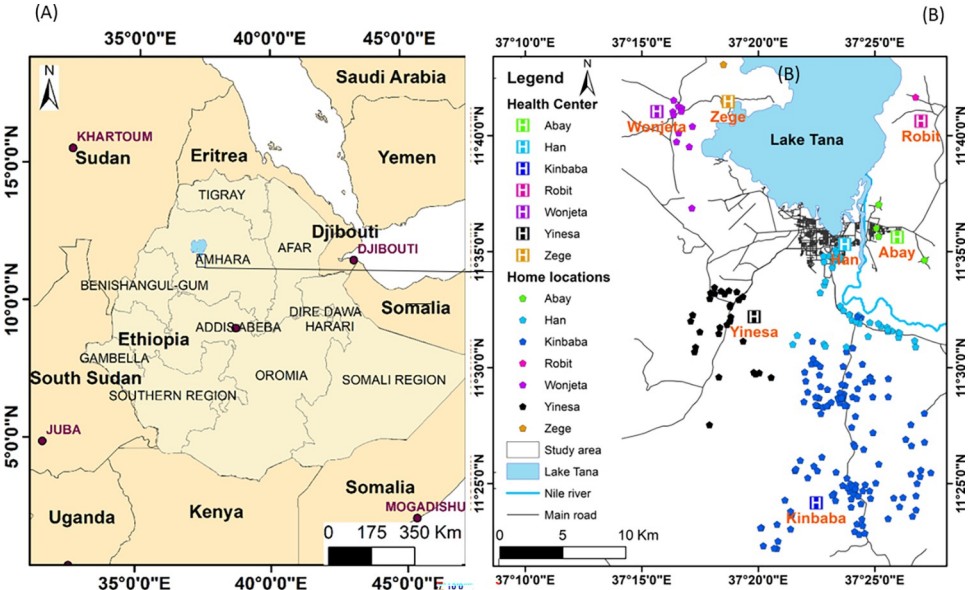

**Fig 1.** (A) Location map of the study area. (B) spatial distribution of health centers and sampled households. N.B: Shapefiles for (A) and (B) are accessed from https://www.naturalearthdata.com/. GPS coordinates of the health centers and households plotted in (B) are collected from each health center and home during sample collection.

received from the administrative offices, Kinbaba, Wonjeta, Yinesa and Robit Health Centers provided health care services for 32,668, 42,898, 17,303 and 28,401 people, respectively. Han, Zege and Abay Health Centers served 9,354, 5,025 and 10,306 people, respectively. From each of the health centers within a 0.4 to 12.8 KM radius, every household with the suspected cases was visited.

### Stool sample and data collection from children

Samples were collected at each health center and 5–10 grams stool per child were transferred in a sterile plastic cup for storage [21], labeled with child code and date of collection. Samples were kept refrigerated at the health centers until transportation to the microbiology laboratory at Bahir Dar University within 4 hours of collection. Along with the sample, data on children's age, sex, residence, and clinical details (fever, abdominal pain, vomiting, duration of diarrhea, and frequency of passing stool per 24 h) were recorded from each child during face-to-face interviews with their parent(s).

### Fecal sample collection from animals

Following the sampling of the children at the health centers, a visit to their respective homes was organized within one week. From the animals present at home, about 25 grams of fresh feces from bovines, sheep, and goats were collected using a clean spoon-attached stool cup [20]. In case of lack of spontaneous defecation from the livestock, feces were collected from the rectum [25]. For poultry, 10–15 grams of fresh cecal droppings were collected. Finally, the family's indoor garden was observed for freshly excreted feces from pets. A combination of pooled and subset sampling methods were employed, tailored to the circumstances of each animal species in various household scenarios. For poultry, bovine, sheep, and goats, pooled or subset sampling were employed, while calves and companion pets were sampled individually, as the latter were only few per household.

### Isolation and preliminary identification of *Campylobacter*

Standard microbiological practices were followed for transportation, isolation, and preservation of isolates. From each stool and fecal sample, about 1 gram was homogenized in 10 ml of sterile physiological water and gently vortexed for 30 seconds. A polycarbonate filter of 0.6-μm pore size (Global Life Sciences Solutions Operations Ltd., Little Chalfont Buckinghamshire, UK) was placed onto a modified charcoal cefoperazone deoxycholate agar plate (*Campylobacter* Blood-Free Selective Agar Base, CM0739, Oxoid, Basingstoke, UK) [26]. Six drops of the fecal suspension were added on top of a filter membrane and allowed to pass passively for 30 minutes at 37˚C, ambient air [27]. After filtration, the filter was removed, and plates were incubated in a glass jar for 48 to 120 h at 37˚C in a microaerobic atmosphere (burning candle) [16,28]. Plates were checked after 48, 72, 96, and 120 h of incubation for the growth of typical *Campylobacter* colonies.

Preliminary identification of *Campylobacter* isolates was made by typical colony appearances on mCCDA plates. Translucent, grey-white with a metallic sheen, watery and spread-like droplets to a continuous film (swarming) colony, were considered presumptive *Campylobacter* colonies [29].

Subsequently, after microscopic confirmation of the presence of fine and spiral rods by safranin staining [14,26], typical colonies were streaked onto a new mCCDA plate, and incubated for 24–72 h at 37˚C, under microaerobic conditions. Performance of mCCDA, at 37˚C, and microaerobic incubation conditions were tested using *Campylobacter* reference strains from the Belgian Coordinated Collections of Microorganisms (BCCM/LMG).

## Storage and transportation of presumptive *Campylobacter* isolates

Upon growth, pure colonies with typical *Campylobacter* morphology were swabbed from the second mCCDA plates with a sterile cotton swab, transferred to cryotubes (Thermo Scientific, Suxhou, Jiangsu, China) with 1 ml sterile sheep blood and stored at—80˚C. An additional aliquot was kept at -20˚C in Eppendorf tubes with 1 ml of sterile physiological water. The presumptive *Campylobacter* isolates were transported on dry ice from Ethiopia to the Microbiology Laboratory (LMG) at Ghent University, Belgium.

## Re-cultivation of presumptive *Campylobacter* isolates

In Belgium, the suspensions were sub-cultured on blood agar (BA) plates (Columbia Agar Base, CM0331, Oxoid, with 5% defibrinated horse blood (E & O Laboratories Limited, Burnhouse, Scotland)) [28,30]. In addition, 0.5 ml of each cell suspension was transferred to 10 ml of Preston Broth (PB) (Nutrient Broth No. 2, CM0067, Oxoid, with 2ml of sterile distilled water and Modified Preston *Campylobacter* Selective Supplement, SR0204E, Oxoid, and 5% lysed horse blood) in a sterile screw-capped bottle [31]. Blood agar plates and PB tubes were incubated at 37˚C in microaerobic conditions through the introduction of a gas generator to provide a mixture of gases (10% CO2, 10% H2, and 80% N2) into metal jars. After 24 h of incubation, 10 μl of each enrichment broth was spread onto a BA plate and incubated for 48 h as described above [26]. The remaining PB tubes were incubated again for 24h. Of samples for which no growth was obtained after 24 h, another 10 μl was spread from the corresponding PB culture onto a BA plate and incubated for 48–120 hrs [31]. Typical *Campylobacter* growth on the BA plates was harvested for further confirmation and identification at the species level using MALDI-TOF MS.

## *Campylobacter* species identification by MALDI-TOF MS

Typical *Campylobacter* colonies from BA plates were analyzed, mass spectra (MSP) generated and identified by MALDI-TOF MS (Bruker's MALDI-TOF Microflex LT/SH LMGE-0422) as described previously [32,33]. Interpretations of results were made by comparing the generated MSP Bruker log scores to the Bruker database MSP-10833 and LM-UGent iD MSP-4778 database using the Bruker Compass software (Bruker Daltonics). To ensure that the run was valid, calibration was performed before each set of measurements using 1μl Bacterial Test Standard (BTS) (Bruker Daltonics). Identification of *Campylobacter* at the species level was considered if scores were ≥ 2.2, according to the Bruker MALDI Biotyper specifications. Isolates that could not be re-cultivated, or yielded a score < 2.2 identified, were further analyzed by PCR.

## PCR-based analysis of genus and species of *Campylobacter*

DNA extraction from isolates was performed using the alkaline lysis method [34], and a *Campylobacter* genus-specific PCR was applied, targeting the 23S rRNA gene using the primer set THERM1 (5'-TATTCCAATACCAACATTAGT-3') and THERM2, (5'-CGGTACGGGCAA-CATTAG-3') (Merck Sigma-Aldrich, Belgium) [35]. The PCR reaction was carried out in a final reaction of 25 μl. Per reaction, 2 μl of the DNA template was used along with PCR buffer (Tris-Cl + KCl + (NH4)2 SO4 + 15mM MgCl2) (QIAGEN), a deoxynucleotide triphosphate (dNTP) mixture (200 μM each), THERM1 primer (0.5 μM), THERM2 primer (0.5 μM), MgCl2 (2 mM) and *Taq* polymerase (0.5 U/μl).

For all amplification experiments, the Veriti™ 96-Well Fast Thermal Cycler (Applied Biosystems) was used with 27 amplification cycles, as described previously [35]. PCR products were size-separated by 1% (w/v) agarose gel electrophoresis in 1% Tris-borate-EDTA (TBE) buffer

(VWR LIFE SCIENCE). Thermophilic *Campylobacter* strains gave amplicons corresponding to 290 bp [35]. The negative control yielded no amplification product.

Samples for which a *Campylobacter* genus-specific amplicon of 290 bp was generated, were subsequently analyzed in a *C. jejuni*/*C. coli* species-specific PCR assay. The primer sets mDmapA1, (5'-CTATTTTATTTTTGAGTGCTTGTG-3') and mDmapA2, (5'-GCTTTAT TTGCCATTTGTTTTATTA-3') were applied targeting the map A gene from *C. jejuni*, and COL3, 5'-AATTGAAAATTGCTCCAACTATG-3' and MDCOL2, 5'-TGATTTTATTATTT GTAGCAGCG-3' (Merck Sigma-Aldrich) targeting the ceuE gene from *C. coli* [36]. The PCR reaction was carried out in a final reaction of 25 μl. Per reaction, 2 μl of the DNA template was used along with PCR buffer (Tris-Cl + KCl + (NH4)2 SO4 + 15mM MgCl2) (QIAGEN), a dNTPs (200 μM each), MgCl2 (2 mM), MDmapA1 (0.5 μM), MDmapA2 (0.5 μM), COL3 (0.5 μM) and MDCol2 0.5 μM) and *Taq* (0.5 U/μl). Amplification reactions were performed using the Veriti™ 96-Well Fast Thermal Cycler (Applied Biosystems™) according to the program cycle described previously [36]. Amplification generated 589 bp and 462 bp DNA fragments for *C. jejuni* and *C. coli*, respectively.

## Antimicrobial susceptibility testing

The antimicrobial susceptibility of *C. jejuni* and *C. coli* to ciprofloxacin, doxycycline, erythromycin, and azithromycin was determined using the Epsilometer gradient strip diffusion method (ETEST) (bioMérieux, Marcy-l'Etoile, France) on Mueller-Hinton Fastidious (MHF) agar (bioMérieux, Marcy-l'Etoile, France). Inocula were prepared by suspending colonies in MHF broth to a 0.5 McFarland density using a nephelometer. Inoculated plates were incubated under micro-aerophilic conditions using a gas mixture of 6% O2, 7.1% CO2, 7.1% H2, and 79.7% N2 (Anoxomat) at 37˚C. After 48 h of incubation, the MIC values were read and interpreted based on the 2023 EUCAST Clinical Breakpoint Tables V.13.0 [23]. *Campylobacter jejuni* ATCC 33560 quality control strain was included for each separate run. Antimicrobials were selected based on availability and prescription in the study area, and standard treatment and AST guidelines. Ciprofloxacin is the first choice of treatment for enteritis in the study area. In addition, ciprofloxacin and azithromycin are listed as first and second choices for enteritis treatment in primary care. All four antimicrobials tested are locally available.

## Statistical analysis

GPS coordinates of the visited homes were recorded, and their spatial distributions were visualized using ArcGIS version 10.5 software. Data were analyzed using IBM SPSS Statistics for Windows, Version 29.0. Armonk, NY: IBM Corp. SYSTAT. Baseline characteristics of the children were summarized using numbers with percentages for categorical variables. Pearson Chi-square tests were used to observe a significant association between categorical variables (gender, different age groups, frequency of stool per day, and duration of diarrhea) and *Campylobacter* infection in children. A Games-Howell post-hoc test was used to identify among which specific age groups the significant difference in *Campylobacter* infection occurs.

The presence of significant association between each symptoms related to diarrhea (watery diarrhea, foul-smelling diarrhea, vomiting, abdominal pain, and fever) and *Campylobacter* infection was evaluated using the chi-square test. Fisher's exact test was used to observe the association between *Campylobacter* infection and bloody stools.

Both the Shapiro-Wilk and Kolmogorov-Smirnov tests were used to test the normality of continuous data such as age of the children in months, duration of diarrhea in days, frequency of diarrhea per day, and number of siblings per household. All these data were not normally distributed and are reported with medians and interquartile ranges (Q1 to Q3). The median

ages of children with and without *Campylobacter* infection were compared using the Mann-Whitney U test. Likewise, the Mann-Whitney U test was used to compare the median duration of diarrhea, the median frequency of diarrhea, as well as the median number of siblings, between children with and without *Campylobacter* infection.

Finally, a chi-square test was computed to assess differences in the rate of resistance to ciprofloxacin between *C. jejuni* and *C. coli* isolates. Likewise, the rate of resistance to doxycycline between *C. jejuni* and *C. coli* was compared using the chi-square test. P-values less than 0.05 were considered statistically significant.

## Results

### Occurrence and identification of *Campylobacter*

As shown in Table 1, 303 children from 303 different households participated in the study and provided a fecal sample. In addition, fecal samples from 253 poultry, 213 bovines, 105 calves (under 1 year of age), 99 sheep, 22 dogs, 17 goats and 2 cats were collected. Upon first screening in Ethiopia (preliminary identification), *Campylobacter* was presumptively present in 250 of the 1014 samples. After shipping to the lab of microbiology, Ghent University, Belgium, presumptive isolates from 34 samples could not be cultivated or identified by PCR. The remaining isolates from 129 samples could be recultivated and resulted in 135 identifications as *Campylobacter* species by MALDI-TOF MS, and an additional 97 identifications via PCR in 87 samples for which re- cultivation was not possible. From these confirmed 216 positive samples, 200 had one type of *Campylobacter* species, and 16 had co-infection with both *C. jejuni* and *C. coli* resulting in the identification of 232 *Campylobacter* isolates. The majority of the isolates were *C. jejuni* (n = 194) followed by *C. coli* (n = 34). Additionally, four animal isolates were identified as *C. fetus* (n = 2) and unspecified *Campylobacter* spp. (n = 2).

In total, 59 of the isolates originated from the stools of 59 children (out of 303 samples), and 173 isolates from 157 animal fecal samples (out of 711 samples). The occurrence of *Campylobacter* in animals varied per host, with rates up to 64% in dogs to 6% in bovines (Table 1). In animals, co-occurrence of *C. jejuni* and *C. coli* was observed in dogs (9%), poultry (4%), sheep (3%), and calves (1%) (Table 1).

The isolation rate of *C. jejuni* was the highest in dogs (59%), followed by poultry (40%), and children (19%), with 8%, 5% and 5% rates in bovines, sheep, and calves, respectively (Table 1).

**Table 1. *Campylobacter* occurrence and species distribution in children, backyard farm animals and companion pets.**

| Number of hosts | *Campylobacter* occurrence | *Campylobacter* species distribution | | | | |
|---|---|---|---|---|---|---|
| | Total | *C. jejuni* | *C. coli* | *C. jejuni* + *C. coli* | *C. fetus* | Unspecified *Campylobacter* spp |
| | N (%) | N (%) | N (%) | N (%) | N (%) | N (%) |
| Children (n = 303) | 59 (20) | 57 (19) | 2 (1) | 0 | 0 | 0 |
| Poultry (n = 253) | 112 (44) | 91 (36) | 11 (4) | 10 (4) | 0 | 0 |
| Bovines (n = 213) | 12 (6) | 10 (5) | 1 (1) | 0 | 1 (1) | 0 |
| Calves (n = 105) | 9 (9) | 4 (4) | 2 (2) | 1 (1) | 1 (1) | 1 (1) |
| Sheep (n = 99) | 9 (9) | 5 (5) | 1 (1) | 3 (3) | 0 | 0 |
| Dogs (n = 22) | 14 (64) | 11 (50) | 1 (5) | 2 (9) | 0 | 0 |
| Goats (n = 17) | 1 (6) | 0 | 0 | 0 | 0 | 1 (6) |
| Cats (n = 2) | 0 | 0 | 0 | 0 | 0 | 0 |
| Total (n = 1014) | 216 (21) | 178 (18) | 18 (2) | 16 (2) | 2 (0.2) | 2 (0.2) |

N: number

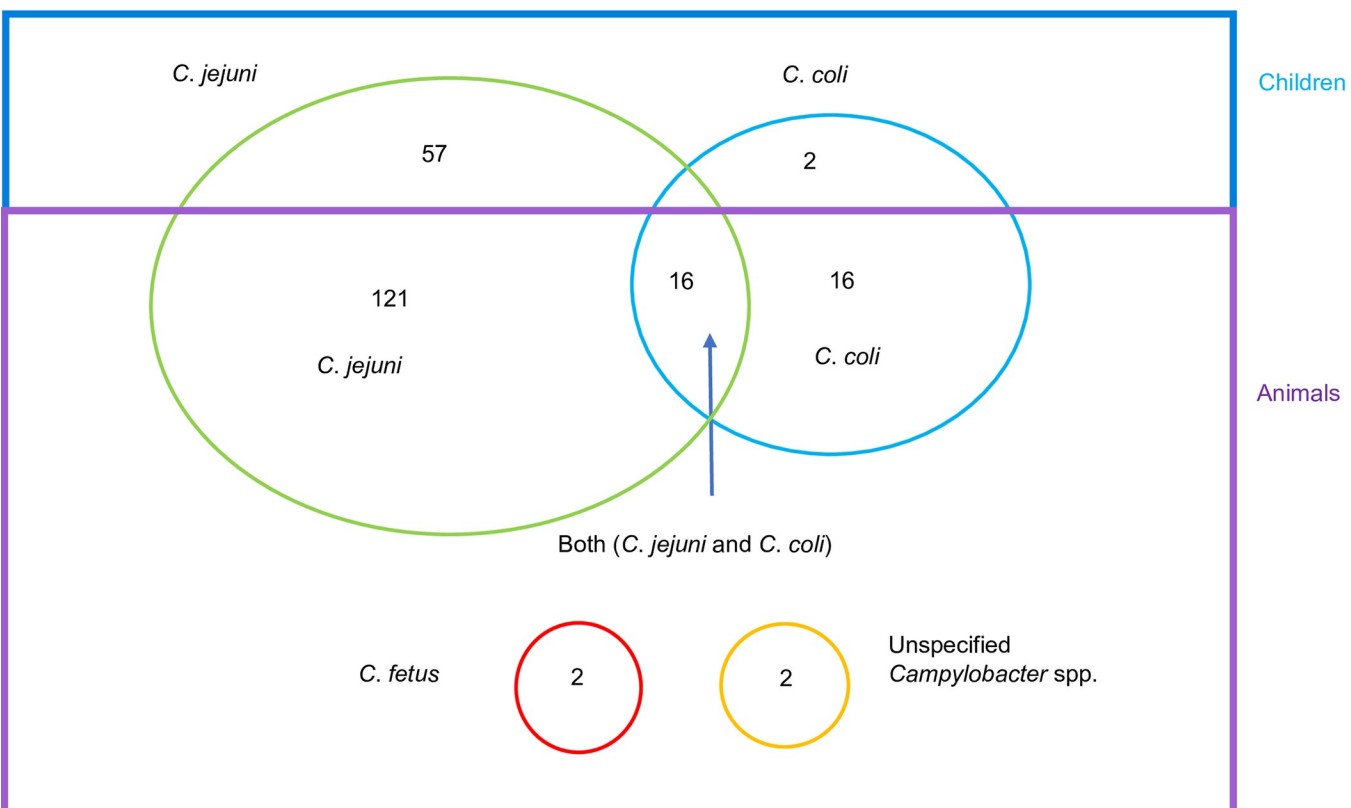

**Fig 2. Prevalence of *Campylobacter* species in children and animals.**

Likewise, *C. coli* was isolated most from dogs (14%), then from poultry (8%), sheep (4%), calves (3%), children (1%), and bovines (1%).

The occurrence of *Campylobacter* is shown in Fig 2. Of the 1014 samples examined, *C. jejuni* was present in 194 (19%) and *C. coli* in 34 (3%), with 16 out of 711 samples from animal origin only (2%), with co-occurrence of *C. jejuni* and *C. coli*. *Campylobacter fetus* was isolated from one bovine and one calf fecal sample. In total at least one *Campylobacter* species was present in 216 of the 1014 samples examined (21%).

## Demographic and clinical details of children with diarrhea and *Campylobacter* infection

Out of the 303 under five children who participated in this study, the exact age of two children missed being recorded. The median age of the 301 children was 19 months with an interquartile range of 12–31.5, and children with *Campylobacter* infection were significantly younger (13 months) compared to those without (19 months) (Mann-Whitney U test: $p < 0.001$).

Based on the Ethiopian Public Health Institute children age group classification [22], the proportion of *Campylobacter* positive cases varied significantly per age group: < 6 months (22%), 6–11 months (38%), 12–23 months (17%), 24–35 months (17%), 36–47 months (9%), and 48–60 months (7%) ($p = 0.002$) (Table 2). Upon post hoc analysis, a significantly higher number of infections were identified in the age groups 6–11 months as compared to 12–23 months ($p = 0.05$), 36–47 months ($p = 0.006$) and 48–60 months ($p = 0.004$) in our cohort (Fig 3).

**Table 2. Demographic and clinical details of children under five with diarrhea with and without *Campylobacter*.**

| Characteristics | Children with *Campylobacter* | Children without *Campylobacter* | Total | p-value |
|---|---|---|---|---|
| **Gender**[#] | | | | |
| Girls | 25 (18) | 115 (82) | 140 (46) | |
| Boys | 34 (21) | 129 (79) | 163 (54) | 0.511 |
| **Age (months)**[#] | | | | |
| < 6 | 2 (22) | 7 (78) | 9 (3) | |
| 6–11 | 21 (38) | 34 (62) | 55 (18) | 0.002 |
| 12–23 | 20 (17) | 101 (83) | 121 (40) | |
| 24–35 | 9 (17) | 44 (83) | 53 (18) | |
| 36–47 | 3 (9) | 32 (91) | 35 (12) | |
| 48–60 | 2 (7) | 26 (93) | 28 (9) | |
| Median (Q1, Q3) | 13 [10, 22.8] | 19 [13, 33.75] | 19 [12, 31.5] | < 0.001 |
| **Frequency of stool per day**[#] | | | | |
| 3–5 | 48 (16) | 200 (66) | 248 (82) | |
| 6–10 | 11 (20) | 44 (80) | 55 (18) | 0.913 |
| Median (Q1, Q3) | 4 [3, 5] | 4 [3, 5] | 4 [3, 5] | 0.538 |
| **Duration of diarrhea (days)**[#] | | | | |
| 1–2 | 18 (16) | 97 (84) | 115 (38) | |
| 3–5 | 33 (24) | 106 (76) | 139 (46) | 0.224 |
| 6–7 | 8 (16) | 41 (84) | 49 (16) | |
| Median (Q1, Q3) | 3 [2, 5] | 3 [2, 4] | 3 [2, 4] | 0.242 |
| **Watery diarrhea**[#] | | | | |
| Yes | 40 (20) | 158 (79) | 198 (65) | 0.659 |
| No | 19 (18) | 86 (82) | 105 (35) | |
| **Foul smelling diarrhea**[#] | | | | |
| Yes | 37 (20) | 148 (80) | 185 (61) | 0.728 |
| No | 22 (19) | 96 (81) | 118 (39) | |
| **Bloody stools**[#] | | | | |
| Yes | 4 (14) | 25 (86) | 29 (10) | 0.417 |
| No | 55 (20) | 219 (72) | 274 (90) | |
| **Vomiting**[#] | | | | |
| Yes | 26 (17) | 124 (83) | 150 (50) | 0.352 |
| No | 33 (22) | 120 (78) | 153 (51) | |
| **Abdominal pain**[#] | | | | |
| Yes | 36 (22) | 130 (78) | 166 (55) | 0.284 |
| No | 23 (17) | 114 (83) | 137 (45) | |
| **Fever**[#] | | | | |
| Yes | 22 (21) | 82 (79) | 104 (34) | 0.593 |
| No | 37 (19) | 162 (81) | 199 (66) | |

[#]Number (%); Q1, Q3, lower quartile, upper quartile

In our cohort, a total of 163 children (54%) were boys, and 140 (46%) were girls. A non-significant higher occurrence of *Campylobacter* was observed in boys (n = 34; 21%) as compared to girls (n = 25; 18%) (Table 2). The median (Q1, Q3) number of siblings, irrespective of their diarrheal status, was 3(2–4). Children with *Campylobacter* infection had a median (Q1, Q3) of 2.5 siblings (1.75–4), compared to 3 (2–4) for those without. The difference was not statistically significant (Mann-Whitney U test: p = 0.26). The duration of diarrhea upon sampling was 3 to

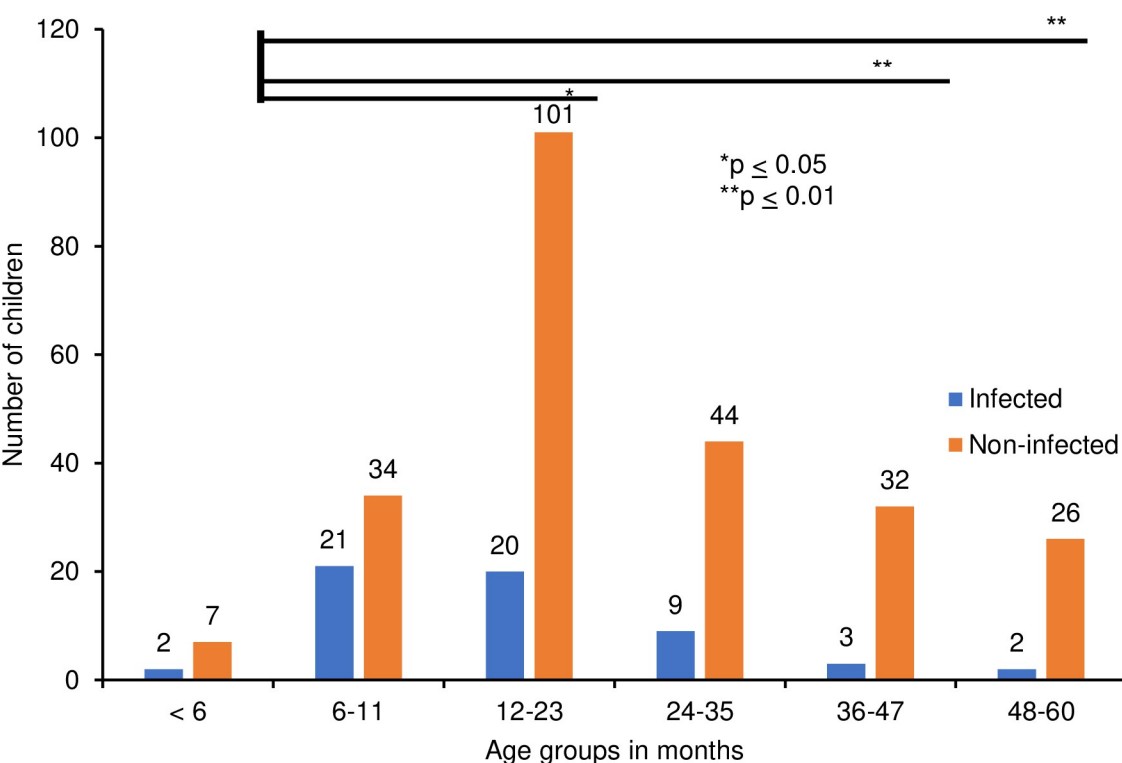

**Fig 3. *Campylobacter* infection per age group of children.** Based on Games-Howell post hoc test multiple comparisons, the proportion of *Campylobacter* infection was significantly higher in age groups from 6 to 11 months compared to 12 to 23 months (p = 0.05), 36 to 47 months (p = 0.006) and 48 to 60 months (p = 0.004). No significant difference was observed between the age-group 6 to 11 months and the remaining age-groups (p > 0.05).

5 days in 46% of children participating. The median duration of diarrhea at presentation was 3 days in both children with *Campylobacter* in their stool and those without. Likewise, the median stool frequency at presentation was 4 times per day in both groups. Watery diarrhea, abdominal pain, vomiting, and bloody stool were reported in 65%, 55%, 50%, and 10% of the children, respectively, with no significant association to *Campylobacter* infection (Table 2).

## Occurrence of *Campylobacter* at the household level

Out of 303 households, 245 households could be visited for the subsequent collection of 711 additional animal fecal samples. The median (Q1, Q3) number of animals included per household was 3 (2, 4). *Campylobacter* was present in at least one host in 149 (61%) of 245 households. In 27 (18%) of these, *Campylobacter* was only detected in the child, while in 23 (15%) it was detected in both a child and an animal species (Fig 4A). *Campylobacter* was detected in a child and poultry in 14 of these 23 households, as well as in child-bovine, child-dog-poultry, child-sheep, child-bovine-dog, and child-calf-poultry combinations in 2, 2, 1, 1, and 1 households, respectively (Fig 4B). The majority (19/23) of these co-occurring identifications were consistent at *Campylobacter* species level between the different hosts. More specifically, 18 were consistent between hosts and were identified as *C. jejuni*, and one identified as *C. coli*. In four cases, different species of *Campylobacter* were identified in different hosts (children and animals) from the same household (Fig 4B).

*Campylobacter* was detected in animal species but not in children in 99 households. Among these, 89 were detected in a single animal species, with 68 (76%) from poultry, 7 (8%) from calves, 5 (6%) from dogs, 5 (6%) from sheep, 3 (3%) from bovines, and 1 (1%) from goats

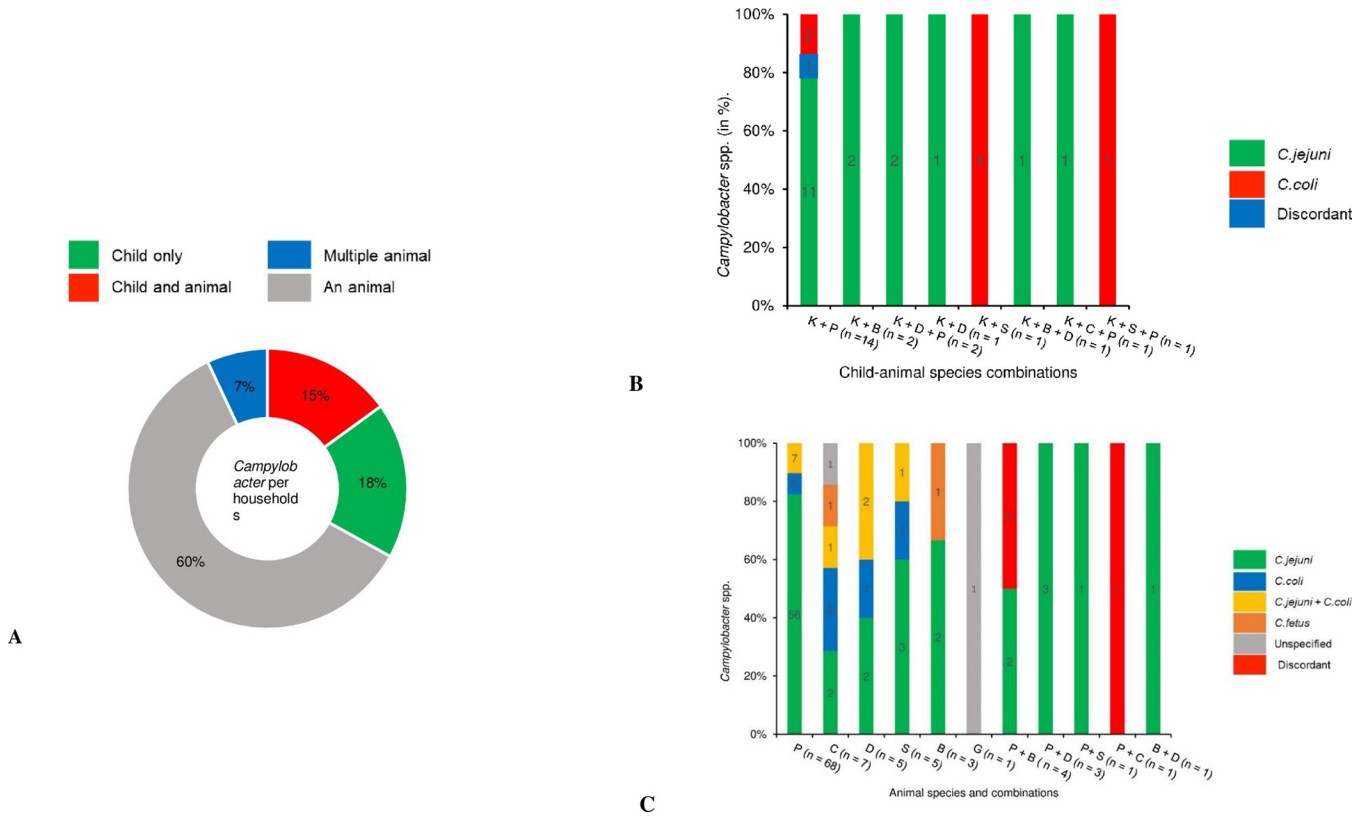

**Fig 4.** Occurrences of *Campylobacter* per 245 households in (A), *Campylobacter* species in child-animals combinations (n = 23) in (B), and per animal species and combinations in (C) (n = 99).

(Fig 4C). *C. jejuni* was responsible for 65 (73%), *C. coli* for 9 (10%), both for 11 (12%), *C. fetus* for 2 (2%), and unidentified *Campylobacter* spp. for 2 (2%) of identifications in the 89 households (Fig 4C). In 10 households, *Campylobacter* occurred in multiple animal species: 4 involved poultry and bovine, 3 poultry and dogs, 1 poultry and sheep, 1 poultry and calf, and 1 bovine and dog. Of these, *C. jejuni* was identified in all hosts in seven households, while in three households a variety of *Campylobacter* species was identified (Fig 4C). Co-occurrence of both *C. jejuni* and *C. coli* in one host was observed exclusively in animal hosts in 9% (14/149) of households and mainly occurred in poultry (8 cases; 57%).

## Spatial distribution of *Campylobacter*

As visualized in Fig 5, the majority of households sampled were from Kinbaba and Han health Center catchment areas. Households positive for *Campylobacter* are observed across the radius of each health center, with a higher concentration in Kinbaba, followed by Han. For children, *Campylobacter* positivity was concentrated in the Kinbaba and Han catchments, while positivity in poultry seemed evenly distributed across the radius of each health center (Fig 5A). At species level, *C. jejuni* exhibited an even distribution, while *C. coli* appeared scattered. A single instance of *C. fetus* was detected in Han and Yinesa (Fig 5B).

## Antimicrobial resistance profiles in *Campylobacter*

As shown in Table 3, MIC values were read and interpreted for 128 *Campylobacter* isolates (103 *C. jejuni*, 24 *C. coli*, and one *C. fetus*). Isolates demonstrated resistance rates of 20% and

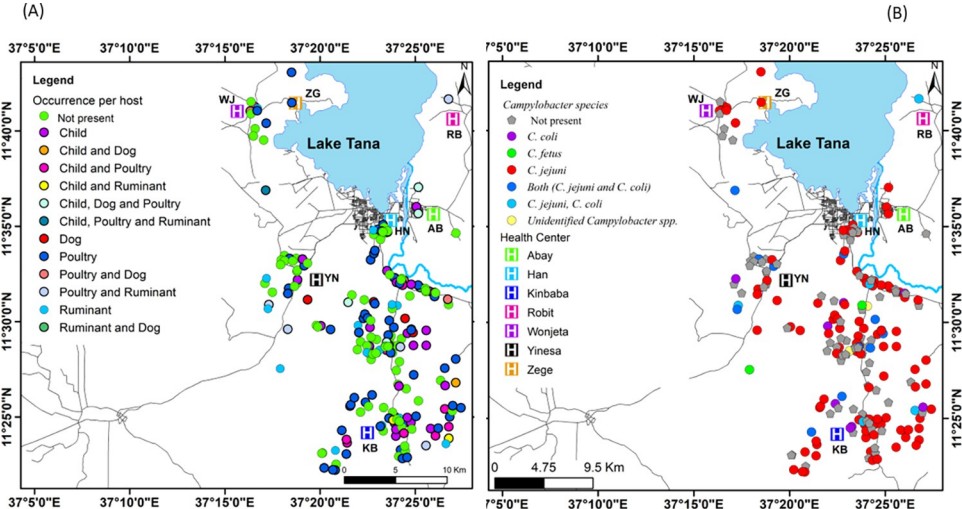

**Fig 5. Spatial distribution of households with and without *Campylobacter*.** Spatial patterns of *Campylobacter* per host in (A). Spatial patterns of *Campylobacter* species per sampled households in (B). N.B: Shapefiles for both (A) and (B) are accessed from https://www.naturalearthdata.com/. GPS coordinates of the health centers and the households plotted are collected from each health center and home during sample collection. In panel (A), households delineated in green indicate those that tested negative for *Campylobacter*, while other colours represent households that tested positive per host. In panel B, grey delineated households indicate those that tested negative for *Campylobacter*, whereas different colours represent positive households per *Campylobacter* species type.

11% to ciprofloxacin and doxycycline, respectively. *C. coli* had higher ciprofloxacin resistance (50%) than *C. jejuni* (14%) (p < 0.001). Resistance to doxycycline was higher in *C. coli* (21%) compared to *C. jejuni* (9%) (p = 0.088).

The proportion of *C. jejuni* isolates resistant to ciprofloxacin was 3 (9%) in children and 10 (19%) in poultry. The proportion of *C. coli* isolates resistant to ciprofloxacin was 3 (75%) in ruminants and 9 (47%) in poultry. All *C. coli* isolates (n = 5) resistant to doxycycline were from poultry. Of the 128 isolates tested, only 1 (1%) from a child was resistant to azithromycin and 1 (1%) from poultry to erythromycin (Table 3).

Overall, (10/128, 8%) co-resistance to ciprofloxacin and doxycycline was obtained. The proportion of co-resistance was significantly higher in *C. coli* (n = 5, 21%) than in *C. jejuni* (n = 6, 6%) (p = 0.019). The proportion of co-resistance to ciprofloxacin and doxycycline was (n = 5, 5%) and (n = 5, 21%) in *C. jejuni* and *C. coli* isolates, respectively. All *C. coli* isolates and the majority of *C. jejuni* isolates resistant to both ciprofloxacin and doxycycline were observed in poultry.

The MICs of ciprofloxacin ranged from 0.008 to > 32 mg/L in *C. coli* and from 0.002 to > 32 mg/L in *C. jejuni*. The MICs of doxycycline ranged from 0.03 to 64 mg/L in *C. coli*, and from

0.016 to 256 mg/L in *C. jejuni*. Ciprofloxacin has a bimodal distribution of MICs for *C. coli* isolates (0.008–0.125 mg/L versus 4–32 mg/L). The MICs for doxycycline are also bimodal (Table 4).

As shown in Table 5, of the *C. jejuni* isolates simultaneously isolated from a child and poultry in 4 households, all were concurrently susceptible to azithromycin, doxycycline and erythromycin. However, in one household, a poultry isolate was resistant to doxycycline, and in another household, a poultry isolate was resistant to erythromycin. For ciprofloxacin, two isolates were concurrently susceptible to increased exposure, and two were resistant in poultry but not in children.

**Table 3. Antimicrobial resistance profiles of 103 *C. jejuni*, 24 *C. coli* and one *C. fetus*, isolated from children and animals.**

| Isolates per source | Azithromycin | Erythromycin | Ciprofloxacin | Doxycycline |
|---|---|---|---|---|
| | N (% R) | N (% R) | N (% R) | N (% R) |
| **Children** | | | | |
| *C. jejuni* (n = 35) | 1 (3) | 0 | 3 (9) | 2 (6) |
| *C. coli* (n = 1) | 0 | 0 | 0 | 0 |
| **Poultry** | | | | |
| *C. jejuni* (n = 54) | 0 | 1 (2) | 10 (19) | 7 (13) |
| *C. coli* (n = 19) | 0 | 0 | 9 (47) | 5 (26) |
| **Dogs** | | | | |
| *C. jejuni* (n = 7) | 0 | 0 | 1 (14) | 0 |
| **Ruminants** | | | | |
| *C. jejuni* (n = 7) | 0 | 0 | 0 | 0 |
| *C. coli* (n = 4) | 0 | 0 | 3 (75) | 0 |
| *C. fetus* (n = 1) | 0 | 0 | 0 | 0 |
| **Total *C. jejuni*** (n = 103) | 1 (1) | 1 (1) | 14 (14) | 9 (9) |
| **Total *C. coli*** (n = 24) | 0 | 0 | 12 (50) | 5 (21) |
| **Overall (n = 128)** | **1 (1)** | **1 (1)** | **26 (20)** | **14 (11)** |

N: number of resistant isolates; % R, percentage of resistance

In two households where *C. jejuni* was isolated from dogs and poultry, isolates from both sources were susceptible to erythromycin and azithromycin. However, in one household, an isolate from a poultry was resistant to ciprofloxacin, and in another household, a poultry isolate was resistant to doxycycline (Table 5). Furthermore, from three households, where *C. jejuni* and *C. coli* were isolated from a single poultry sample, both species were susceptible to all antibiotics. Similarly, from a single calf in a household, both *C. jejuni* and *C. coli* isolated were susceptible to all antibiotics (Table 5).

**Table 4. Distribution of antimicrobial susceptibility results to four antimicrobials of 103 *C. jejuni* and 24 *C. coli*, isolated from children with diarrhea and backyard farm animals.**

**Number of isolates with the following MIC (mg/L)**

| Clinical breakpoint for resistance (mg/L) | | 0.002 | 0.004 | 0.008 | 0.016 | 0.03 | 0.06 | 0.125 | 0.25 | 0.5 | 1 | 2 | 4 | 8 | 16 | 32 | 48 | 64 | 256 |
|---|---|---|---|---|---|---|---|---|---|---|---|---|---|---|---|---|---|---|---|
| **Azithromycin** *C. jejuni* | > 4 | | | | | 12 | 37 | 42 | 9 | 1 | | | 1 | | | | | | 1 |
| *C. coli* | > 8 | | | | | | 2 | 3 | 8 | 6 | 5 | | | | | | | | |
| **Ciprofloxacin** | | | | | | | | | | | | | | | | | | | |
| *C. jejuni* | > 0.5 | 3 | | 3 | 14 | 22 | 26 | 17 | 2 | 2 | | 1 | | 1 | | 12 | | | |
| *C. coli* | > 0.5 | | | | 1 | 1 | 5 | 4 | 1 | | | | 1 | 1 | | 10 | | | |
| **Doxycycline** | | | | | | | | | | | | | | | | | | | |
| *C. jejuni* | > 2 | | | | | 52 | 33 | 2 | 5 | 1 | | 1 | | 3 | 3 | | | 1 | 2 |
| *C. coli* | > 2 | | | | | 16 | 3 | | | | | | | | | 1 | | 4 | |
| **Erythromycin** *C. jejuni* | > 4 | | | | | 3 | 14 | 28 | 24 | 17 | 11 | 3 | 1 | 1 | | | | 1 | |
| *C. coli* | > 8 | | | | | | 2 | 5 | 8 | | 9 | | | | | | | | |

MIC, minimum inhibitory concentration; the grey shading indicates MIC values of the resistant isolates

**Table 5. Antimicrobial susceptibility profiles of concurrent *Campylobacter* species isolated from the child and animal species, and from distinct animal species per home.**

| *Campylobacter* positive home number | Hosts per home | *Campylobacter* species | Antimicrobials tested | | | |
|---|---|---|---|---|---|---|
| | | | Erythromycin | Ciprofloxacin | Azithromycin | Doxycycline |
| | **A child and animal species** | | | | | |
| 90 | Child | *C. jejuni* | S | I | S | S |
| | Poultry | *C. jejuni* | R | R | S | S |
| 139 | Child | *C. jejuni* | S | I | S | S |
| | Poultry | *C. jejuni* | S | I | S | S |
| 239 | Child | *C. jejuni* | S | I | S | S |
| | Poultry | *C. jejuni* | S | R | S | S |
| 256 | Child | *C. jejuni* | S | I | S | S |
| | Dog | *C. jejuni* | S | I | S | S |
| 288 | Child | *C. jejuni* | S | I | S | S |
| | Poultry | *C. jejuni* | S | I | S | R |
| | **Different animal species** | | | | | |
| 117 | Dog | *C. jejuni* | S | I | S | S |
| | Poultry | *C. jejuni* | S | R | S | S |
| 280 | Dog1 | *C. jejuni* | S | I | S | S |
| | Dog2 | *C. jejuni* | S | I | S | S |
| | Poultry1 | *C. jejuni* | S | I | S | R |
| | Poultry2 | *C. jejuni* | S | I | S | S |
| | **A single animal species** | | | | | |
| 108 | Poultry | *C. coli* | S | I | S | S |
| | | *C. jejuni* | S | I | S | S |
| 124 | Poultry | *C. coli* | S | I | S | S |
| | | *C. jejuni* | S | I | S | S |
| 196 | Calf | *C. coli* | S | I | S | S |
| | | *C. jejuni* | S | I | S | S |
| 205 | Poultry | *C. coli* | S | I | S | S |
| | | *C. jejuni* | S | I | S | S |

S, Susceptible; R, Resistant: I, Susceptible to increased exposure

## Discussion

This study presents the occurrence, spatial distribution, and AMR of *Campylobacter* in a large cohort of under five children with acute diarrhea and various animal species. Using a membrane filtration culture technique and candle extinction microaerobic incubation system, we successfully isolated thermotolerant *Campylobacter*. MALDI-TOF MS and PCR confirmed 86% of presumptive isolates as *Campylobacter*. Compared to earlier studies in Ethiopia, the rate of infection in children is high, with children aged 6 to 11 months having the highest infection rates. Co-occurrences of *Campylobacter* in both children and animals sharing the same home may suggest possible household-level transmission. The spatial distribution suggests that the vicinity of Bahir Dar may be a hotspot area for the presence of *Campylobacter*. Antimicrobial resistance profiles showed low resistance to erythromycin and azithromycin but higher rates for ciprofloxacin and doxycycline, particularly in poultry. The findings could be applicable to regions with similar socio-economic and cultural contexts, where backyard farming is common.

The fastidious nature of *Campylobacter* challenges the reliability of stool culture for estimation of the burden of *Campylobacter* in regions with restricted resources for laboratories.

Previous studies in Ethiopia lacked consistency in culture techniques impeding a comprehensive understanding of *Campylobacter* epidemiology. To address this gap, the current study, the first of its kind in Ethiopia, adopted a membrane filtration culture technique that resulted in the successful isolation of thermotolerant *Campylobacter* from children, backyard farm animals and dogs. A significant portion of presumptive samples were confirmed as *Campylobacter* via molecular tools.

Out of the 232 *Campylobacter* identifications, 58% were identified using MALDI-TOF MS, and 42% via PCR. The cells which could not be resuscitated and recovered by culture but that were identified via PCR, could be the viable but non-culturable (VBNC) cells [37], or injured cells due to freeze-thaw during an unstable power supply, or the lack of representation in the MALDI- TOF MS database. Moreover, presumptive isolates in a subset of samples not identified as *Campylobacter* by molecular tools could be bacteria such as *Proteus* and *Acinetobacter*, which resemble *Campylobacter* morphology on mCCDA media [38], or emerging *Campylobacter* species.

Compared to earlier studies in different regions of Ethiopia, the rate of *Campylobacter* infection in children (20%) in the present study, exceeded the rates reported in southwest (9–17%) [39,40], southern (7–13%) [41,42], and northwest (6–15%) [43–45] Ethiopia. Previous studies in Ethiopia typically used selective *Campylobacter* agar without filtration, and incubation at 42˚C for 24 to 48 hours. A study in eastern Ethiopia reported a detection rate of 15% in infants by PCR [7]. However, another study from southern Ethiopia [21], reported a detection rate of 49% in infants using selective chromogenic agars. The rate of infection in children is also higher than previously reported rates in Africa [8], Asian countries [46,47], Ecuador [48] and Poland [49]. The variation in the findings could be due to the stool culture techniques used to isolate and identify *Campylobacter*. The studies in Africa [8,39,40,42,44,45] applied biochemical tests that have identification limitations stemming from the low metabolic activity of *Campylobacter* [30]. Thus, the membrane filtration culture method, combined with specific incubation conditions is an effective and easy method for screening *C. jejuni*/*C. coli* in routine diagnosis, and research in settings lacking organized laboratories.

Differences in recruitment methods for including diarrheic children could also contribute to variations in *Campylobacter* occurrence. A study in northwest Ethiopia [43] recruited diarrheic children through house-to-house searches, whereas the present study enrolled children seeking medical care for diarrheal illness. This is because many symptomatic instances of *Campylobacter* might not seek medical attention. Differences in frequency and duration of diarrhea, admission status, and considerations for immunodeficiency and malnutrition across studies, could contribute to variations in *Campylobacter* occurrence among children. In the present study, children with a minimum of three episodes of diarrhea per day lasting for 1 to 7 days were included, while studies in northwest Ethiopia [43,45] enrolled those with watery diarrhea once daily for 1 to more than 14 days. Studies from Asia enrolled admitted children [46,47], whereas outpatient children were included in the present study. Additionally, a study in Lebanon [46] excluded children with immunodeficiency, malnutrition, and multiple malformations, a factor not accounted for the present study. Furthermore, children from urban and rural settings may have different transmission intensities of *Campylobacter*. Factors such as differences in sanitation and hygiene levels and nutrition between studies could contribute to observed variations in the occurrence of *Campylobacter* in children.

In the present study, the number of *Campylobacter* infections was significantly higher in children aged 6 to 11 months compared to those aged 36 to 47 months and 48 to 60 months in line with previous studies in Ethiopia [45], and elsewhere in the world [8,50,51]. As reported in Ethiopia [39,40,45], higher infection rates were observed in children below three years old in the present study (p = 0.012). This can be attributed to differences in the development and

maturation of intestinal microbiota among the different age groups [52]. The intestinal microbiota of the children is more mature from the age of two years onwards [53,54]. Earlier exposure to *Campylobacter* also induces specific partially protective immunity. Besides, children from 6 to 11 months are more likely to be exposed to contaminated surfaces, foods, and water after weaning, and have less developed immunity [8,52].

In our study, *C. jejuni* prevailed *C. coli* in both children and animal species consistent with what is reported before [19]. This could be due to the diverse nature of *C. jejuni*, enabling its widespread colonization in mammals and poultry [55]. Moreover, *C. jejuni* has better virulence factors and adaptation mechanisms that enable it to survive and proliferate in the human gastrointestinal tract compared to *C. coli* [55]. The observed occurrence of *C. jejuni* in children (19%) in the present study exceeded previously documented rates in Ethiopia (9 to 12%) [40,56] as well as rates reported among children in the African context (5.2–12.3%) [8]. The rate of *C. jejuni* in ruminants (5–8%) is comparable with previous reports from other parts of Ethiopia (ranging from 3% to 9%) [7,18,57]. A higher rate of *C. jejuni* has already been reported from the feces of poultry (81%) and cattle (54%) in southwest Ethiopia [58]. Methodological differences could contribute to the observed variations, as several earlier studies in Ethiopia relied on biochemical tests [18,41,57,58].

The overall rate of *C. coli* (3%) in the present study fell within the spectrum of rates (ranging from 1% to 11%) reported in previous studies mentioned above [18,20,40,56,58]. The isolation rate of *Campylobacter coli* was higher in animal species than in children in the present study. The strains of *C. coli* prevalent in animals might be less adapted to infect and cause disease in children. Additionally, the simultaneous excretion of both *C. jejuni* and *C. coli* was observed in backyard farm animals and companion pets but not in children. This could be due to competition with the intestinal microbiota, and variations in exposure to pathogens, body temperature, strain virulence, and immune responses [26,59].

Apart from a study in central Ethiopia reporting an 8% prevalence of *C. fetus* from livestock [20], other studies in Ethiopia did not isolate *C. fetus* from fecal animal samples. The use of an incubation temperature of 42° C inhibits the growth of most non-*C. jejuni* and non-*C. coli Campylobacter* spp. The presence of *C. fetus* only in one bovine and one calf stool sample and the inability to isolate other slow-growing *Campylobacter* species in the present study might be linked to the absence of hydrogen gas in the candle extinction microaerobic conditions during isolation [27,60].

Regarding animal species, the occurrence of *Campylobacter* spp. was higher in dogs and poultry compared to sheep and cattle in concurrence with a finding from southwest Ethiopia where the percentage of isolation was higher in chicken compared to sheep and cattle [58]. Conversely, other studies in the central and southwest regions of Ethiopia reported higher rates of *C. jejuni* in ruminants and *C. coli* in sheep [20,58,61]. The isolation rate of *Campylobacter* among household dogs (64%) was higher compared to rates from other countries (11–45.4%) [62,63], possibly due to the unrestricted movements of dogs in the neighborhood, potentially increasing their exposure to various sources of contamination, including the excreta of children and from consumption of dairy products [64]. Similarly, the isolation rate of *Campylobacter* in poultry (44%) was higher than a report from central Ethiopia (13%) [20]. Relatively lower rates of *Campylobacter* isolation in sheep (9%) and cattle (6–9%) were documented compared to previous reports in Ethiopia (11%- 38% in sheep and 12.6–12.7% in cattle) [18,57,58]. However, considering the fact that 85% of households in Ethiopia own backyard farm animals [22], and keep them in family houses, and the concurrent excretion of *C. jejuni* and *C. coli* in a few cattle in the current study highlights the need for further research. Differences in animal species, age, and handling practices and geographic regions may be

attributed to the observed differences in *Campylobacter* occurrences among backyard farm animals.

Investigating *Campylobacter* occurrence between children and animals sharing the same house is crucial for designing targeted interventions. *Campylobacter* was detected in 61% of the family's homes visited, which is higher than a previous study from central Ethiopia (42%) [20]. This could be due to variations in the age and diarrhea status of human participants. We screened stool samples collected only from children under five with diarrhea seeking medical care at health centers, while the study in central Ethiopia collected pooled human fecal samples at the household level regardless of both diarrheal status and age specifications.

The occurrence of joint cases of *C. jejuni* among children and animals (for instance, child-poultry in 14 houses, child-bovines in 2, and child-dog-poultry in 2 homes), along with the matched AST profiles of *C. jejuni* from children and animals in two homes, may suggest the presence of possible household-level transmission of *Campylobacter* via exposure to floors and various surfaces contaminated with feces. This is strengthened by the free access of animals to the sitting, kitchen, and dining areas of the families, as observed during data collection. This finding is in accordance with other studies [20,48,65]. However, the detection of *Campylobacter* exclusively in children in some homes or exclusively in animals in other homes, as well as the unmatched AST profiles of *C. jejuni* isolates from children and animals in three homes, could indicate the presence of different contamination sources [65].These indicate the complexity of the transmission dynamics of *Campylobacter*.

Knowledge on the AMR of *Campylobacter* isolates from children, backyard farm animals, and companion pets is scarce in Ethiopia. The present study determined, for the first time in Ethiopia, MICs for the antimicrobials recommended to treat human campylobacteriosis. Only 0.8% of tested *C. jejuni* isolates were resistant to erythromycin and azithromycin, which is consistent with recent CDC, European Union (EU), and global reports [49,66,67]. However, a MIC of azithromycin ($\geq$ 256 mg/l) was observed in a *C. jejuni* from a child. This could be due to the unrestricted use of azithromycin for mild pneumonia in the study area. Higher rates of resistance to erythromycin were reported in *Campylobacter* isolates from young children in southwest Ethiopia (11–18%) [39,40], and northwest Ethiopia (28%) [44] and in poultry in central Ethiopia (15%) [20]. These studies employed disc diffusion techniques for susceptibility testing.

The 20% rate of ciprofloxacin resistance is consistent with findings in Europe (18–22%) [59,68]. However, variable resistance rates to ciprofloxacin were reported in isolates from children in the southwest (5.3% to 15.8%) [39,40], southern (55%) [41] and northwest (16%) [44] Ethiopia. An 8% rate of resistance in bovines and 13% combined (human and backyard farm animals) were also reported in Ethiopia [18,20]. *C. coli* displayed significantly higher resistance to ciprofloxacin than *C. jejuni* (p < 0.001), consistent with reports from EU [67]. The high rates of ciprofloxacin resistance among *C. jejuni* isolates from poultry and *C. coli* isolates from calves (67%) in the present study is concurrent with a study from Poland for *C. jejuni* [69] and a report from EU countries for *C. coli* [67]. As only a few isolates from ruminants were tested in this study, caution needs to be taken in the interpretation of this finding. The MIC of ciprofloxacin showed a bimodal distribution in *C. coli* compared to a monomodal one in *C. jejuni* indicating different resistance mechanisms or the presence of subpopulations in *C. coli*. The MICs for doxycycline are also bimodal and in agreement with the EUCAST epidemiological cut-offs (ECOFFs).

The overall rate of *Campylobacter* resistance to doxycycline (11%) was lower compared to worldwide reports (28–42%) [70,71]. The rate of resistance was considerably higher in *C. coli* than in *C. jejuni*, consistent with reports from the EU [67]. The rate of doxycycline resistance in *C. jejuni* was higher among isolates from poultry than children consistent with the study

from Poland [69]. The 8% ciprofloxacin-doxycycline co-resistance observed in *Campylobacter* isolates in the present study is in line with the 6% and 9% rates reported in U.S., [72] but lower than the 15% co-resistance rates in Bulgaria [68]. Almost 90% of ciprofloxacin-doxycycline co-resistance was observed among isolates from poultry. The overall co-resistance was significantly higher in *C. coli* isolates than *C. jejuni*, consistent with the report from EU [67].

This study presents the matching between susceptibility profiles of isolates from children and animals (Table 5). The presence of matched AST profiles in two homes was confirmed, while unmatched in three homes. In general, the resistance of *Campylobacter* to ciprofloxacin and doxycycline in the present study may be due to the common use of these antimicrobials for treating enteric infections in the study area. The predominance of resistance in isolates from backyard poultry where *Campylobacter* can easily multiply, survive in droppings for up to 6 days after excretion [73] and spread to children, humans, and companion pets is of particular concern. The observed resistance in poultry may be due to specific poultry-associated mechanisms, reverse zoonotic transmission, acquired resistance from wild birds, local farming, or environmental sources, rather than selection pressure, as antimicrobials are not used in backyard poultry in this particular setting. Moreover, the unmatched AST profiles of *C. jejuni* isolates across poultry and children, and poultry and dogs observed in a few homes (Table 5) support these explanations.

Assessment of the occurrence of *Campylobacter* using large sample sizes at the children-animal interface, the use of membrane filtration culture technique for the first time in Ethiopia, as well as successful transportation from Ethiopia to Belgium on dry ice, and subsequent recultivation under microaerobic conditions by introducing a gas mixture, were the strengths of this study. In addition, the confirmation of isolates using MALDI-TOF MS and determination of AMR based on gradient strip diffusion susceptibility testing is the first for *Campylobacter* from Ethiopia.

There are also several important limitations that should be considered when interpreting findings from this analysis. The present study does not use enrichment before filtration-based plating, which may have limited the isolation of *Campylobacter* that were present in low numbers or under stress in the tested matrix. While the method employed favorably isolated *C. jejuni* and *C. coli*, which are responsible for the majority of gastrointestinal diseases related to *Campylobacter* infection, it may have overlooked slow-growing emerging *Campylobacter* species such as *C. lari*, *C. hominis*, *C. upsaliensis*, *C. hyointestinalis*, *C. curvus*, *C. concisus*, and others due to the lack of a hydrogen-enriched atmosphere during the initial isolation microaerobic incubation. Additionally, fast-growing *Campylobacter* may also be missed in some samples. Furthermore, the use of candle extinction microaerobic incubation may have a lower isolation rate compared to methods involving a gas generator to provide a mixture of gases (10% $CO_2$, 10% $H_2$, and 80% $N_2$).

Moreover, the cross-sectional nature of the study and the lack of genetic analysis limit to elucidate the transmission dynamics of *Campylobacter* between animal species and children, even in households where *C. jejuni* co-occurs among both. Additionally, due to logistical constraints and feasibility, environmental samples such as surfaces, dust from the living areas of children and animals, and water sources were not examined. This narrows the discussion on the role of the environment in transmission or as an opportunity for intervention. As a result, the present study primarily addresses the human and veterinary domains of the One Health doctrine.

## Conclusions

*Campylobacter jejuni* is prevalent in children below the age of 3 years with diarrhea in Ethiopia. Its abundance in children, shedding from poultry and companion pets and concordant

presence in both a child and animal species with matching AST profiles in a few homes may suggest possible household-level transmission between animals and children. Based on the findings, treatment of severe campylobacteriosis in children should be guided by E-test susceptibility testing. In order to preserve efficacy and minimize the risk of AMR development, azithromycin and erythromycin should only be used for the treatment of severe campylobacteriosis that is resistant to ciprofloxacin. The observed high prevalence of *Campylobacter* in children and co-occurrences among children and animals and animal species within households suggest further genetic analysis and longitudinal studies to comprehensively understand *Campylobacter* transmission, mechanism of AMR, and ascertain causality to children. This finding highlights the need for targeted interventions such as raising awareness of transmission routes, improving hygiene, and proper handling of animals and disposal of their waste. Clinicians, veterinarians, and policymakers should integrate these findings for evidence-based diagnosis and treatment, further research, surveillance, and mitigation of *Campylobacter* transmission.

## Supporting information

**S1 Table. Demographic data vs. Campylobacter infection in children.**
(XLSX)

**S2 Table. Campylobacter occurrence per hosts included.**
(XLSX)

**S3 Table. Antimicrobial Susceptibility test result.**
(XLSX)

**S4 Table. Campylobacter_AST_MIC values.**
(XLSX)

**S1 Data. Data for the map creation.**
(RAR)

## Acknowledgments

Special thanks to LM-UGent for hosting the laboratory analysis. Special thanks to Margo Cnockaert, Martine Boonaert and Carine van Lancker for providing training and excellent technical assistance. The authors gratefully acknowledge the technical skills and support of the technicians of the microbiology laboratory in Saint-Lucas Hospital. We are very grateful to the children and their parents who participated in the study. Thanks to local health administrations for their crucial support and facilitation of the fieldwork during the Covid-19 pandemic and wartime in Ethiopia. The authors would like to thank the Microbiology staff of the Amhara Public Health Institute and Felegehiwot Referral Hospital for providing sterile sheep blood. We extend our thanks to the healthcare staff of the study health centers for their assistance in data and stool sample collection. We thank community workers, and volunteers for their unreserved help and coordination of the fieldwork. Special thanks to Mr. Aynenew Abate and Mr. Belaynew Kerisew, and the local motorcyclists for their community awareness and support in the house-to-house data collection. We extend our gratitude to Dr. Ashebir Sewale Belay for his invaluable assistance in mapping the study area and spatial distributions.

## Author Contributions

**Conceptualization:** Wondemagegn Mulu, Marie Joossens, Mulugeta Kibret, Kurt Houf.

**Data curation:** Wondemagegn Mulu, Kurt Houf.

**Formal analysis:** Wondemagegn Mulu, Marie Joossens, Kurt Houf.

**Funding acquisition:** Marie Joossens, Mulugeta Kibret, Kurt Houf.

**Investigation:** Wondemagegn Mulu, Marie Joossens, Anne-Marie Van den Abeele, Kurt Houf.

**Methodology:** Wondemagegn Mulu, Marie Joossens, Mulugeta Kibret, Anne-Marie Van den Abeele, Kurt Houf.

**Project administration:** Wondemagegn Mulu, Marie Joossens, Mulugeta Kibret, Kurt Houf.

**Resources:** Marie Joossens, Mulugeta Kibret, Kurt Houf.

**Software:** Kurt Houf.

**Supervision:** Mulugeta Kibret, Kurt Houf.

**Validation:** Wondemagegn Mulu, Marie Joossens, Kurt Houf.

**Visualization:** Wondemagegn Mulu, Kurt Houf.

**Writing – original draft:** Wondemagegn Mulu, Marie Joossens, Kurt Houf.

**Writing – review & editing:** Wondemagegn Mulu, Kurt Houf.

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
