## [Decision Letter · Decision Letter 0]

16 Mar 2024

Dear Prof. dr. Houf,

Thank you very much for submitting your manuscript "Campylobacter in children under the age of five with diarrhea, backyard farm animals, and companion animals: Occurrence and antibiotic resistance" for consideration at PLOS Neglected Tropical Diseases. As with all papers reviewed by the journal, your manuscript was reviewed by members of the editorial board and by several independent reviewers. The reviewers appreciated the attention to an important topic. Based on the reviews, we are likely to accept this manuscript for publication, providing that you modify the manuscript according to the review recommendations. 

Sincerely,

Kirkby D Tickell, MBBS BSc

Academic Editor

Elsio Wunder Jr

Section Editor

Reviewer's Responses to Questions

**Key Review Criteria Required for Acceptance?**

**Methods**

-Are the objectives of the study clearly articulated with a clear testable hypothesis stated?

-Is the study design appropriate to address the stated objectives?

-Is the population clearly described and appropriate for the hypothesis being tested?

-Is the sample size sufficient to ensure adequate power to address the hypothesis being tested?

-Were correct statistical analysis used to support conclusions?

-Are there concerns about ethical or regulatory requirements being met?

Reviewer #1: The goal of this study, to use new methods to quantify Campylobacter in young children experiencing diarrhoeal disease and in animals in Ethiopia is well articulated. The methods seem appropriate to address this objective and the populations which were sampled are well described. However, it is unclear if all animals in the home were sampled or if only a subset were sampled. If the data is available, information about children who were not eligible to participate due to insufficient sample collection would be useful to include as well.

Reviewer #2: Methodologically, the manuscript sounds good to address all the issues mentioned above.

**Results**

-Does the analysis presented match the analysis plan?

-Are the results clearly and completely presented?

-Are the figures (Tables, Images) of sufficient quality for clarity?

Reviewer #1: The analysis matches the methods described. The structure of Table 1, with the different sections for occurrence and identification could be more clearly presented, as in the current form the difference between the two sections is not immediately clear. Overall the description of the results is comprehensive. However, in the section on demographic data in children, it is mentioned that there is no association between infection and number of siblings but no additional information is provided in Table 2 or in the text. In Table 5 is is also unclear why these specific sample pairs were selected to be presented. Based on Figure 4, they do not appear to be all instances of these concurrent isolations.

Reviewer #2: The results are nicely presented using figures, graphics, and narrative methods. Both on the document itself and in a separate file, there are a few little problems stated.

**Conclusions**

-Are the conclusions supported by the data presented?

-Are the limitations of analysis clearly described?

-Do the authors discuss how these data can be helpful to advance our understanding of the topic under study?

-Is public health relevance addressed?

Reviewer #1: The authors conclude that these results suggest direct animal to human transmission. However, given the cross-sectional nature of this study, concluding transmission instead of simply co-occurrence in both groups is a not supported by the data. In addition, the unmatched AST profiles between animals and children support the possibility of other explanations, which the authors do mention later in the discussion. Discussion of limitations is minimal and could be further expanded upon to include limitations other than those related to molecular methods. The authors discuss their findings in the context of previous work done both in Ethiopia and globally. However, further consideration of reasons for differences in findings beyond culture techniques could be discussed if they exist. In the one study where this is discussed, the differences in methods with regards to who was sampled are significant, which raises the question of the inclusion criteria in other studies which are discussed. Finally, while Campylobacter infection is arguably a One Health issue, and this study makes reference to One Health at several points, this study does not include all three domains of One Health. It does not include environmental samples and includes minimal discussion of the role of the environment in transmission or as an opportunity for intervention.

Reviewer #2: To emphasize the importance of the One Health doctrine, the work approach is relatively new in the study location (Ethiopia).

**Editorial and Data Presentation Modifications?**

Reviewer #1: In the results section on child demographics, reference is made to as group classifications of the Ethiopian Public Health Institute. However, the phrasing of this significant difference is unclear and could be reworded or clarified. The first sentence in the discussion also states that the study presents the spatial distribution of Campylobater. However, while the spatial distribution of sampled households is presented, those that were positive are not delineated.

Reviewer #2: Minor revision.

The detail is organized on a separate file.

**Summary and General Comments**

Reviewer #1: In general, this study presents useful information about the burden of Campylobacter in the study region. It's strengths include it's methods and the number of samples, as well as the description of the distribution and characteristic of positive samples. The authors adequately discuss these findings in the context of previous findings. However, the conclusions drawn from the study regarding transmission are ambitious given the cross sectional nature of the study and lack of genetic analysis. The limitations of this study and how it fits into a One Health approach could also be expanded. In addition, there are places where the writing could be reworded to improve clarity.

Reviewer #2: The detail is organized on a separate file.

PLOS authors have the option to publish the peer review history of their article (what does this mean?). If published, this will include your full peer review and any attached files.

Reviewer #1: No

Reviewer #2: No

Figure Files:

Data Requirements:

Reproducibility:

References

---

## [Decision Letter · Decision Letter 1]

22 May 2024

Dear Prof. dr. Houf,

We are pleased to inform you that your manuscript 'Campylobacter occurrence and antimicrobial resistance profile in under five-year-old diarrheal children, backyard farm animals, and companion pets' has been provisionally accepted for publication in PLOS Neglected Tropical Diseases.

Best regards,

Kirkby D Tickell, MBBS BSc

Academic Editor

Elsio Wunder Jr

Section Editor

Reviewer's Responses to Questions

**Key Review Criteria Required for Acceptance?**

**Methods**

-Are the objectives of the study clearly articulated with a clear testable hypothesis stated?

-Is the study design appropriate to address the stated objectives?

-Is the population clearly described and appropriate for the hypothesis being tested?

-Is the sample size sufficient to ensure adequate power to address the hypothesis being tested?

-Were correct statistical analysis used to support conclusions?

-Are there concerns about ethical or regulatory requirements being met?

Reviewer #1: (No Response)

**Results**

-Does the analysis presented match the analysis plan?

-Are the results clearly and completely presented?

-Are the figures (Tables, Images) of sufficient quality for clarity?

Reviewer #1: (No Response)

**Conclusions**

-Are the conclusions supported by the data presented?

-Are the limitations of analysis clearly described?

-Do the authors discuss how these data can be helpful to advance our understanding of the topic under study?

-Is public health relevance addressed?

Reviewer #1: (No Response)

**Editorial and Data Presentation Modifications?**

Reviewer #1: (No Response)

**Summary and General Comments**

Reviewer #1: My previously identified concerns have adequately been addressed in this revision.

PLOS authors have the option to publish the peer review history of their article (what does this mean?). If published, this will include your full peer review and any attached files.

Reviewer #1: No

---

## [Editor Report · Acceptance letter]

29 May 2024

Dear Prof. dr. Houf,

We are delighted to inform you that your manuscript, "Campylobacter occurrence and antimicrobial resistance profile in under five-year-old diarrheal children, backyard farm animals, and companion pets," has been formally accepted for publication in PLOS Neglected Tropical Diseases.

Best regards,

Shaden Kamhawi

co-Editor-in-Chief

Paul Brindley

co-Editor-in-Chief
